# Molecular characterization of *Campylobacter* spp. recovered from beef, chicken, lamb and pork products at retail in Australia

Rhiannon L. Wallace[1], Dieter M. Bulach[2,3], Amy V. Jennison[4], Mary Valcanis[3], Angus McLure[1], James J. Smith[5], Trudy Graham[4], Themy Saputra[6], Simon Firestone[7], Sally Symes[8], Natasha Waters[9], Anastasia Stylianopoulos[8], Martyn D. Kirk[1], Kathryn Glass[1]*

**1** National Centre for Epidemiology and Population Health, The Australian National University, Canberra, Australian Capital Territory, Australia, **2** Melbourne Bioinformatics, The University of Melbourne, Melbourne, Victoria, Australia, **3** Microbiological Diagnostic Unit Public Health Laboratory, The Peter Doherty Institute, Melbourne, Victoria, Australia, **4** Public Health Microbiology, Forensic and Scientific Services, Queensland Health, Brisbane, Queensland, Australia, **5** Food Safety Standards and Regulation, Health Protection Branch, Queensland Health, Brisbane, Queensland, Australia, **6** New South Wales Food Authority, NSW Government, Sydney, New South Wales, Australia, **7** Melbourne Veterinary School, Faculty of Veterinary and Agricultural Sciences, The University of Melbourne, Parkville, Victoria, Australia, **8** Department of Health and Human Services, Victoria State Government, Melbourne, Victoria, Australia, **9** ACT Government Analytical Laboratory, Australian Capital Territory Health Directorate, Canberra, Australian Capital Territory, Australia

* kathryn.glass@anu.edu.au

**Data Availability Statement:** The GenBank accession numbers of individual isolates are listed in S1 Table and are available on bioproject PRJNA591966.

## Abstract

Australian rates of campylobacteriosis are among the highest in developed countries, yet only limited work has been done to characterize *Campylobacter* spp. in Australian retail products. We performed whole genome sequencing (WGS) on 331 *C. coli* and 285 *C. jejuni* from retail chicken meat, as well as beef, chicken, lamb and pork offal (organs). *Campylobacter* isolates were highly diverse, with 113 sequence types (STs) including 38 novel STs, identified from 616 isolates. Genomic analysis suggests very low levels (2.3–15.3%) of resistance to aminoglycoside, beta-lactam, fluoroquinolone, macrolide and tetracycline antibiotics. A majority (>90%) of isolates (52/56) possessing the fluoroquinolone resistance-associated T86I mutation in the *gyrA* gene belonged to ST860, ST2083 or ST7323. The 44 pork offal isolates were highly diverse, representing 33 STs (11 novel STs) and harboured genes associated with resistance to aminoglycosides, lincosamides and macrolides not generally found in isolates from other sources. Prevalence of multidrug resistant genotypes was very low (<5%), but ten-fold higher in *C. coli* than *C. jejuni*. This study highlights that *Campylobacter* spp. from retail products in Australia are highly genotypically diverse and important differences in antimicrobial resistance exist between *Campylobacter* species and animal sources.

**Funding:** This work was funded by a National Health and Medical Research Council grant (NHMRC GNT1116294), with partner funding from AgriFutures, Australian Government Department of Health, Food Standards Australia New Zealand, New South Wales Food Authority and Queensland Health. Additional funding was provided by ACT Health. The National Health and Medical Research Council provided research fellowship funding for Martyn D. Kirk (APP1145997). The funders had no role in study design, data collection and analysis, decision to publish, or preparation of the manuscript.

**Competing interests:** The authors whose names are listed immediately below certify that they have NO affiliations with or involvement in any organization or entity with any financial interest (such as honoraria; educational grants; participation in speakers' bureaus; membership, employment, consultancies, stock ownership, or other equity interest; and expert testimony or patent-licensing arrangements), or non-financial interest (such as personal or professional relationships, affiliations, knowledge or beliefs) in the subject matter or materials discussed in this manuscript. Author Names: Rhiannon L. Wallace, Dieter M. Bulach, Amy V. Jennison, Mary Valcanis, Angus McLure, James J. Smith, Trudy Graham, Themy Saputra, Simon Firestone, Sally Symes, Natasha Waters, Anastasia Stylianopoulos, Martyn D. Kirk, Kathryn Glass* AgriFutures is the only commercial entity that provided funding for this study. This does not alter our adherence to PLOS ONE policies on sharing data and materials. All other funding was from NHMRC and various government departments, all of which have now been mentioned in the funding statement.

# Introduction

Thermophilic *Campylobacter coli* and *Campylobacter jejuni* are the most common causes of foodborne bacterial infections worldwide. Campylobacteriosis is also the most frequently notified enteric pathogen under surveillance by OzFoodNet, Australia's government-established foodborne disease surveillance network [1], with 143.5 cases per 100,000 population reported in 2019 [2]. Although only a small fraction of people who become ill from food they have consumed seek medical attention [3], the global burden of foodborne disease is substantial. In Australia, foodborne gastroenteritis is responsible for an estimated 4.1 million cases annually [4]. Globally, the incidence and prevalence of campylobacteriosis have increased dramatically over the past decade [5]. Campylobacteriosis is often self-limiting and generally does not require medical treatment. However, some cases of *Campylobacter* infection are associated with serious clinical manifestations such as bacteraemia, reactive arthritis, haemolytic uremic syndrome, meningitis, septicaemia and Guillain-Barré syndrome [5, 6].

The global overuse and misuse of antimicrobial agents in humans, animals and plants, has greatly accelerated the development of resistance to antimicrobials by pathogenic microorganisms. In the United States alone, the CDC estimates that 2 million people become infected with antimicrobial(s)-resistant bacterial pathogens, resulting in at least 23,000 deaths annually [7]. The development of antimicrobial resistance (AMR) and emergence of multidrug resistant pathogens are global concerns for both public health agencies and the agri-food industry. Antimicrobial resistant pathogens increase the risk of an infected individual suffering an adverse health effect, such as reduced treatment efficacy, and increased disease severity, hospitalization and mortality than individuals infected with an antimicrobial-susceptible pathogen [8–10].

Traditionally, phenotypic-methods have been widely used to characterize *C. coli* and *C. jejuni*. However, molecular techniques, offering greater accuracy and specificity, have replaced phenotypic methods. These techniques include pulsed-field gel electrophoresis (PFGE), multilocus sequence typing (MLST) [11], and more recently next generation sequencing [12]. Whole genome sequencing (WGS) provides the highest possible microbial subtyping resolution available to public health authorities [13], enabling faster detection and identification of resistance determinants/mechanisms in microorganisms [14].

Poultry is Australia's largest meat commodity, with the average Australian consuming 47.4 kg each year [15]. In 2018–2019, 1.24 million tonnes of chicken meat were produced, representing a gross production value of $2.8 billion [16]. The majority of commercial meat chickens in Australia are grown intensively in conventional barns or sheds with controlled environments which provide favourable conditions for the proliferation of pathogenic microorganisms, such as *Campylobacter* spp. Australia has strict regulations regarding antimicrobial use in livestock. Fluoroquinolones, colistin and 4th generation cephalosporins have never been registered for use in Australian food-producing animals, gentamycin use is banned and 3rd generation cephalosporin usage remains restricted [17].

Evaluation of isolates from meat at the retail level using WGS is an effective way to identify the risk of human exposure to enteric pathogens, particularly microorganisms harbouring AMR genes. Despite the size of the animal agriculture industry in Australia, little is known about the AMR profiles of foodborne pathogens, such as *C. coli* and *C. jejuni* in retail products. The aim of this study was to apply WGS to (i) determine the dominant sequence types (STs), (ii) identify the AMR genotypes and (iii) characterize the genetic relatedness of *C. coli* and *C. jejuni* from beef, chicken, lamb and pork products at retail in Australia.

## Materials and methods

### Sample collection and bacterial isolation

Raw meat and offal products were collected from retail outlets in New South Wales (NSW), Queensland (QLD) and Victoria (VIC), between March 2017 and March 2019, as previously described [18]. Additional samples were collected from retail outlets in the Australian Capital Territory (ACT) over a period of five months (May-Sept. 2018). Briefly, fresh and frozen chicken meat and beef-, chicken-, lamb-, and pork-offal were collected from local supermarkets and butcher shops: Canberra in the ACT; Hunter region and metropolitan suburbs around Sydney in NSW; Brisbane, Toowoomba, Rockhampton, Townsville and Cairns in QLD; and Bendigo and Melbourne in VIC. Samples of chicken meat included whole bird, and breast, drumstick, Maryland (thigh and drumstick), thigh, and wing portions. Samples of beef, chicken, lamb and pork offal included giblets, heart, liver, kidney, neck and tongue. *Campylobacter* spp. prevalence on beef, lamb and pork meat is generally low. Therefore organs (offal) were sampled in order to obtain a suitable number of isolates for WGS.

*C. coli* and *C. jejuni* were isolated and identified in their respective jurisdiction according to ISO 10272–1:2017 [19] and AS 5013.06.2015 [20] with minor modifications [18]. Briefly, meat or offal samples were combined with buffered peptone water or enrichment broth (Bolton or Preston) and agitated manually. Samples were incubated at 37°C for 2–4 h followed by micro-aerobic incubation (85% $N_2$, 5% $O_2$, and 10% $CO_2$) at 41.5°C for 44 ± 4 h and subsequently identified to genus and species level as described previously [18].

### Genomic DNA extraction and whole genome sequencing

Genomic DNA was extracted from *C. coli* (*n* = 331) and *C. jejuni* (*n* = 285) isolated from beef, chicken, lamb and pork using the QiaSymphony® DSP DNA Mini kit (Qiagen) according to the manufacturer's protocol. The Nextera XT DNA Library Prep kit (Illumina, San Diego, CA, USA) was used to prepare DNA for sequencing. WGS was performed on the Illumina Next-Seq500 with 150 base-pair paired-end reads using the NextSeq 500 Mid Output kit (300 cycles) (Illumina). Table 1 summarizes the number of isolates sequenced from each sample type.

### Bioinformatic analysis

Paired-end sequences were analysed using the Nullarbor pipeline v2 (https://github.com/tseemann/nullarbor). Read quality was evaluated to ensure sufficient depth of coverage (min 50x) and isolate purity using Kraken (lack of contaminating reads) (https://github.com/DerrickWood/kraken). Sequences were trimmed using Trimmomatic v0.38 to remove Illumina Nextera adapters and low-quality sequences. Isolates with a genome size (total assembled bases) that differed by >20% from the average genome size in the analysis group were

**Table 1. Summary of whole genome sequenced *Campylobacter* isolates, collected from retail outlets in Australia over a period of two years (2017–2019), in this study.**

| Sample type | Year | | | Total |
|---|---|---|---|---|
| | 2017 | 2018 | 2019 | |
| Beef offal | 18 | 8 | 3 | **29** |
| Chicken meat | 167 | 160 | 19 | **346** |
| Chicken offal | 69 | 62 | 7 | **138** |
| Lamb offal | 28 | 29 | 2 | **59** |
| Pork Offal | 12 | 29 | 3 | **44** |
| **Total** | **294** | **288** | **34** | **616** |

excluded. Reads were aligned to a reference genome NC_022132 (*C. coli*) or NC_003912 (*C. jejuni*) using Snippy v4.3.6/BWA-MEM v0.7.17-r1188. Sequences were *de novo* assembled into contigs using SPAdes v3.13.0 and annotated using Prokka v1.13.3 (https://github.com/tseemann/prokka), as part of the Nullarbor pipeline.

*In silico* multilocus sequence typing (MLST) was performed on *de novo* assembled contigs using a BLAST-based tool (https://github.com/tseemann/mlst) using the PubMLST "*Campylobacter jejuni/coli*" allele database (https://pubmlst.org/campylobacter/) sited at the University of Oxford [21].

## Identification of genetic determinants of resistance

Assembled contigs were screened for known AMR genes using the NCBI's AMRFinderPlus (https://www.ncbi.nlm.nih.gov/pathogens/antimicrobial-resistance/AMRFinder/) and Abricate v.0.8.12 (https://github.com/tseemann/abricate). Quality of hits were filtered using a cut-off set at 95% nucleic acid sequence identity and 50% sequence coverage. Mutations in two housekeeping genes, *gyrA* (T→I at amino acid at position 86) and 23S rRNA (nucleotides at position 2074 and 2075), associated with quinolone [22] and macrolide/lincosamide/ketolide [23] resistance, respectively, were investigated. A mutation in the promoter region of the *bla*$_{OXA-61}$ gene (G→T at position 57), which inhibits transcription resulting in the isolate becoming sensitive to ampicillin, despite the presence of the gene [24] was also investigated. BLAST [25] was used to extract the respective nucleotide sequences associated with the 23S rRNA and *bla*$_{OXA-61}$ genes and multiple sequence alignments were generated using Clustal-Omega [26]. The GyrA amino acid sequence was used as the query for a tblastn to extract input sequences for a multiple alignment of the GyrA protein from each isolate.

Temporal analysis of AMR in *C. coli* and *C. jejuni* was assessed by classifying isolates as resistant based on the presence of resistance genes or mutations, which have been shown to have a high concordance with phenotypic resistance [14, 27, 28]. Isolates were grouped by quarter and year based on the date the sample was collected from the retail outlet. If an isolate had one or more of the following (i) resistance genes, *aad9*, *aadE-Cc*, *ant(6)-la*, *aph(3')-llla*, *bla*$_{OXA-184}$, *bla*$_{OXA-185}$, *lnu*(C), *tet*(O) or *tet*(44) or (ii) mutations, *bla*$_{OXA-61}$ G57T, *gyrA* T86I or 23S rRNA A2075G, they were classified as resistant. The prevalence of resistant isolates was determined by dividing the number of isolates with at least one resistance marker in each sampling quarter by the total number of isolates collected in that quarter. Ninety-five percent confidence intervals (CI$_{95}$) for the prevalence of resistant isolates were calculated using the binom package [29] in R [30] with ggplot2 [31] used to plot associated results.

In order to identify resistance-associated factors, a multivariable logistic regression analysis was performed using the *glm* function in R, and the questionr package [32] was used to calculate the odds ratios (ORs) and CI$_{95}$. Isolates were classified as resistant as described previously. Isolates that did not possess any resistance genes or mutations were classified as sensitive. Variables in the model included year of sample collection, region (state or territory), food source and *Campylobacter* species.

## Phylogenetic analysis

Phylogenetic trees were inferred from single nucleotide polymorphisms (SNPs) within the core genome of 331 *C. coli* and 285 *C. jejuni*. FastTree v.2.1.10 [33] with the Jukes-Cantor model for building approximation of maximum phylogenetic tree based on core genome (regions of reference genomes to which reads mapped from each of the isolates in the species-level analysis groups). Interactive tree of life (iTOL) v4 was used for visualization [34]. PHY-LOViZ Online [35] was used to generate goeBURST minimum spanning trees, based on the

seven gene MLST allele profiles, in order to examine the population structure of *C. coli* and *C. jejuni*. Clonal complexes were determined by grouping multilocus genotypes that shared four or more identical alleles among the seven loci (*aspA*, *ginA*, *gltA*, *glyA*, *pgm*, *tkt* and *uncA*) with at least one other genotype in the group.

### Nucleotide sequence accession numbers

WGS results of the 616 *Campylobacter* isolates used in this study were submitted to the National Center for Biotechnology Information (NCBI) (https://www.ncbi.nlm.nih.gov/). The GenBank accession numbers of individual isolates are listed in S1 Table and are available on bioproject PRJNA591966. Note: sequences from 617 isolates are available in this bioproject, including one *Campylobacter hyointestinalis* isolate not discussed in this publication.

## Results

### ST prevalence by source

The 331 *C. coli* isolates differentiated into 60 STs and of these, 25 were observed for the first time in this study (Fig 1). The STs of four of the *C. coli* isolates could not be determined (S1

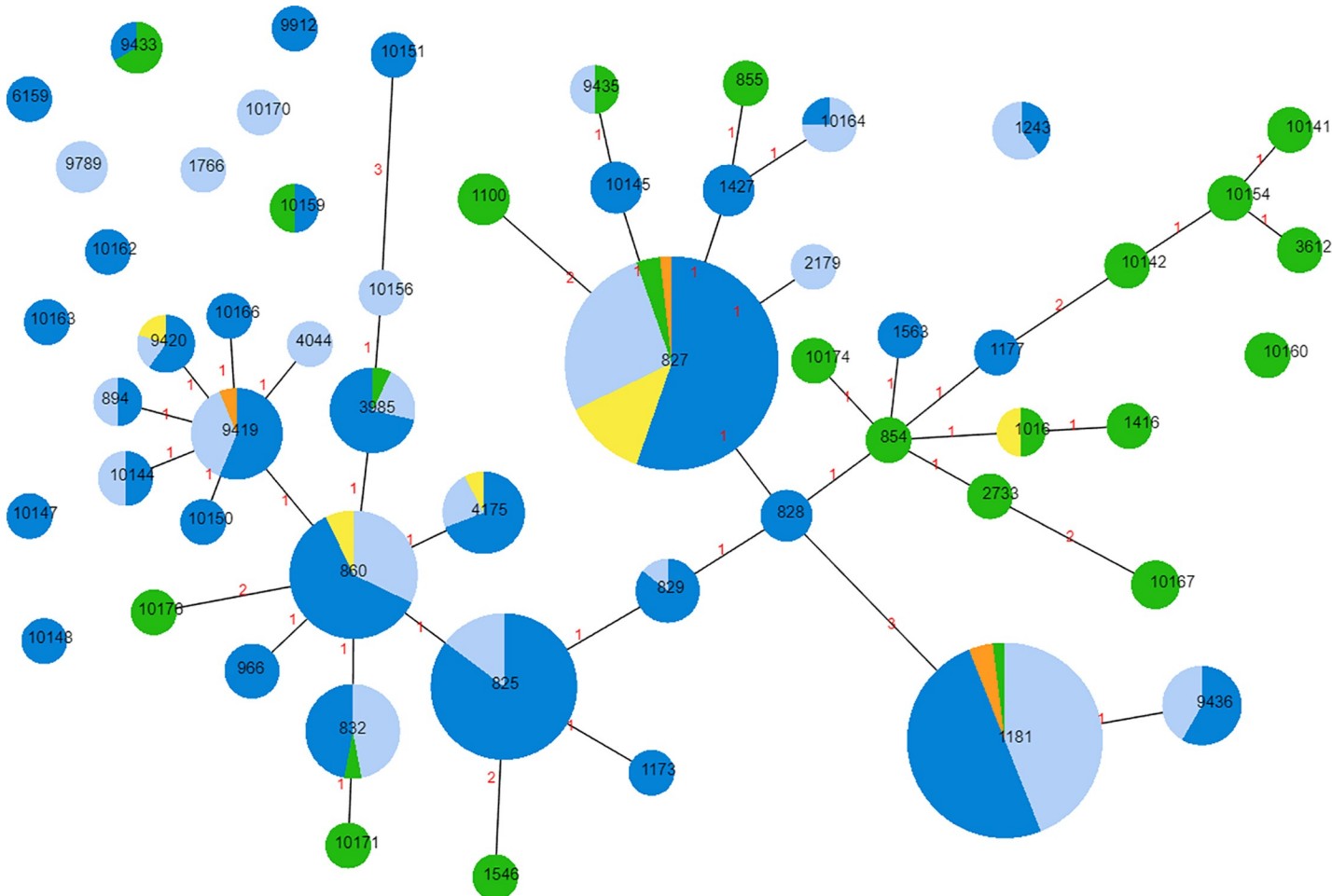

**Fig 1. Distribution of *C. coli* (*n* = 327) Sequence Types (STs) in Australian retail products.** Chicken meat (dark blue), chicken offal (light blue), lamb offal (yellow), beef offal (orange) and pork offal (green). Increasing circle size represents a larger number of isolates of the respective ST. Connecting lines infer phylogenetic relatedness and represent STs with four or more loci in common. Red numbers indicate the number of alleles differing between the two adjoining STs.

Table). Among isolates where the ST was determined, 94.5% belonged to a single clonal complex (ST828 complex), which was dominated by chicken isolates at the core and pork isolates in the periphery. Among *C. coli* isolates, the most frequently observed STs were ST827 (*n* = 56; 16.9%), ST1181 (*n* = 50; 15.1%), ST825 (*n* = 34; 10.3%) and ST860 (*n* = 28; 8.4%). The four most common STs included isolates from at least two animal sources, except ST825 which represented isolates from chicken meat or offal only (*n* = 34). *C. coli* isolates from pork were widely distributed among STs, with 28 isolates representing 23 STs. A majority of the pork isolates (*n* = 18; 64.3%) represented 15 STs not found in any other source.

The 285 *C. jejuni* isolates differentiated into 53 STs and among these, 13 novel STs were identified (Fig 2). In contrast to the population structure of *C. coli*, the population structure of

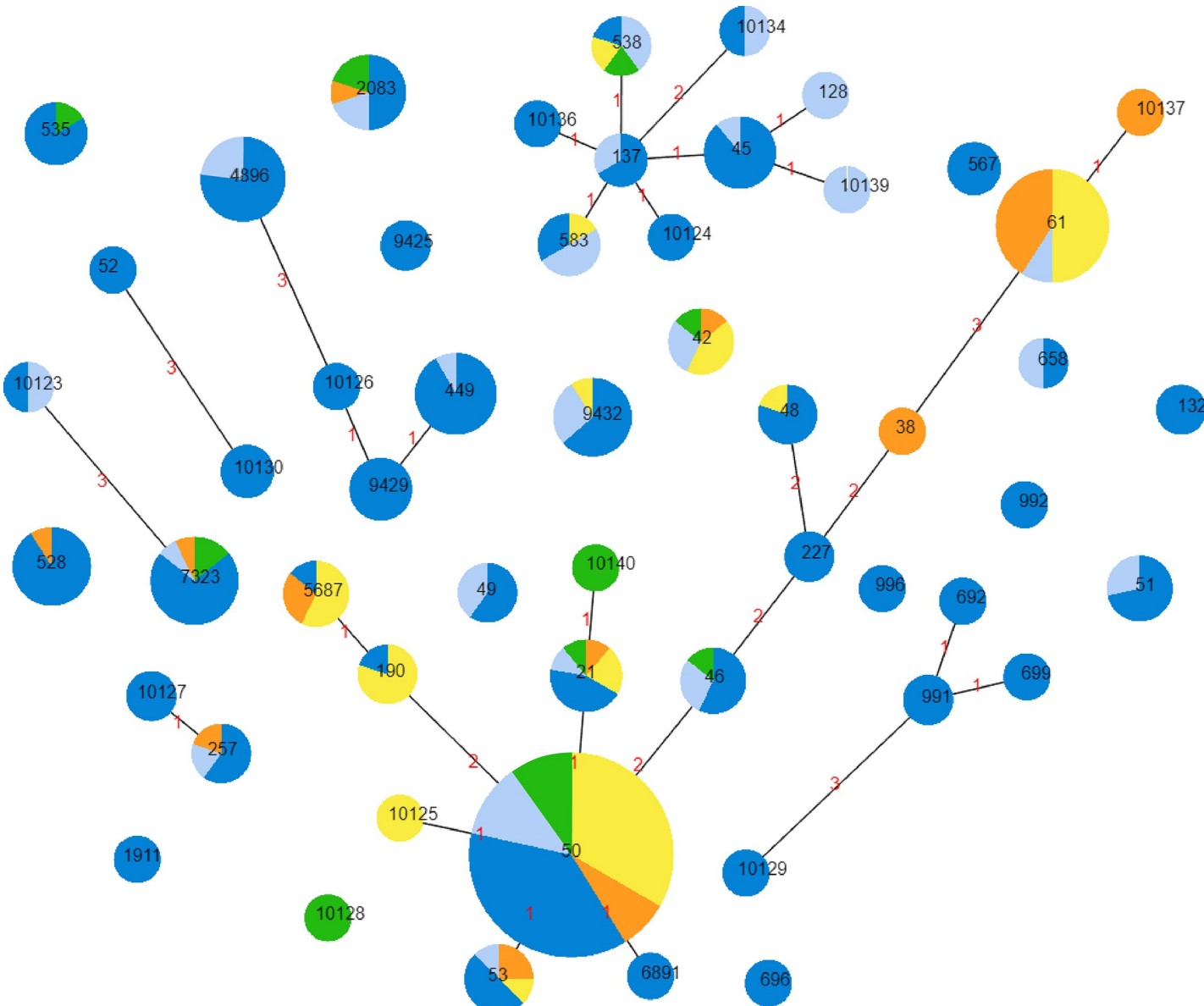

**Fig 2. Distribution of *C. jejuni* (*n* = 285) Sequence Types (STs) in Australian retail products.** Chicken meat (dark blue), chicken offal (light blue), lamb offal (yellow), beef offal (orange) and pork offal (green). Increasing circle size represents a larger number of isolates of that respective ST. Connecting lines infer phylogenetic relatedness and represent STs that have four or more loci in common. Red numbers indicate the number of alleles differing between the two adjoining STs.

*C. jejuni* was more fragmented, with the majority of isolates (*n* = 214; 75.1%) belonging to one of seven clonal complexes. Among *C. jejuni* isolates, ST50 (*n* = 51; 17.9%) and ST61 (*n* = 22; 7.7%) were the most common. Only ST50, ST21 and ST42 contained *C. jejuni* isolates from all four animal sources. ST61 was dominated by isolates from non-chicken sources, namely lamb (*n* = 11) and beef (*n* = 9) offal. Although the *C. jejuni* isolates were predominantly from chicken meat or offal (*n* = 197; 69.1%), 10 STs were dominated (>50%) by isolates from non-chicken sources.

### AMR prevalence over time

The prevalence of resistance genes and point mutations known to confer phenotypic resistance to aminoglycoside, β-lactam, lincosamide, quinolone and tetracycline antibiotics, in *C. coli* and *C. jejuni* collected over two years is shown in Fig 3. Resistance was not analysed by sampling quarter and animal source due to the small number of isolates obtained from non-chicken sources (S1 Fig). Genes and mutations associated with antibiotic resistance were found in all sampling quarters. The prevalence of resistance genes and mutations was lowest in 2017 (6.3–24.3%) and highest in 2018 (27.7–57.6%).

Three variables: year, source and species were associated with either an increased or decreased prevalence of AMR at a significance level of $P < 0.05$ (Table 2). Compared to 2017, isolates collected in 2018 were associated with in an increased risk of resistance (OR 2.41; $CI_{95}$ 1.58–3.68). *Campylobacter* spp. from pork offal were associated with an increased risk of resistance compared to chicken meat (OR 3.59; $CI_{95}$ 1.79–7.37). Among the two *Campylobacter*

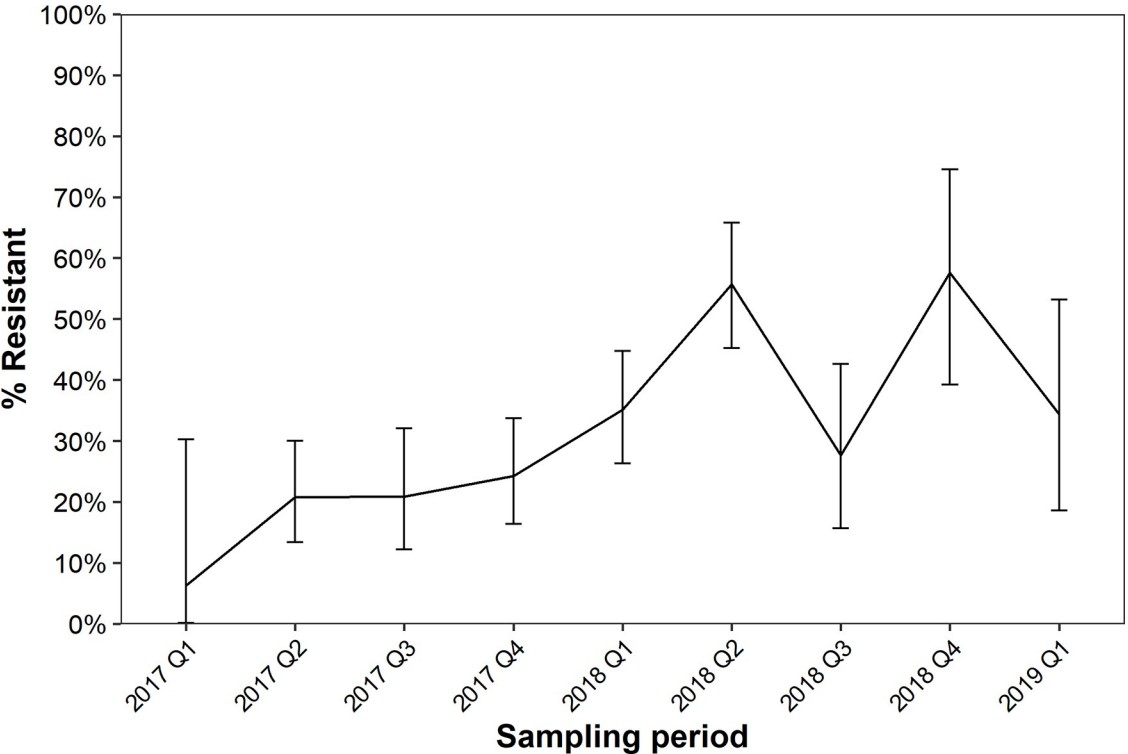

**Fig 3. Temporal analysis of antimicrobial resistance determinants.** Prevalence of *Campylobacter* (*n* = 616) possessing at least one genetic marker of antimicrobial resistance from retail beef, chicken, lamb and pork products collected over a period of two years. Genetic determinants of resistance used to classify isolates as resistant are described in section 2.4. Error bars indicate 95% confidence intervals.

**Table 2. Multivariable logistic regression analysis of factors potentially associated with resistance in *Campylobacter coli* and *Campylobacter jejuni* isolated from retail meat and offal products in Australia, 2017–2019.**

|  | *n* = 616 | OR | CI$_{95}$ | *P* value |
|---|---|---|---|---|
| **Year (reference = 2017; *n* = 294)** |  |  |  |  |
| **2018** | 288 | 2.41 | 1.58–3.68 | <0.000 |
| **2019** | 34 | 1.77 | 0.75–4.02 | 0.178 |
| **Region (reference = Australian Capital Territory; *n* = 69)** |  |  |  |  |
| New South Wales | 212 | 1.04 | 0.55–2.01 | 0.896 |
| Queensland | 188 | 0.70 | 0.35–1.40 | 0.311 |
| Victoria | 147 | 1.56 | 0.84–2.94 | 0.160 |
| **Source (reference = Chicken meat; *n* = 346)** |  |  |  |  |
| Beef offal | 29 | 0.39 | 0.09–1.21 | 0.147 |
| Chicken offal | 138 | 0.94 | 0.60–1.46 | 0.778 |
| Lamb offal | 59 | 0.57 | 0.25–1.19 | 0.154 |
| Pork offal | 44 | 3.59 | 1.79–7.37 | <0.000 |
| ***Campylobacter* species (reference = *C. jejuni*; *n* = 285)** |  |  |  |  |
| *C. coli* | 331 | 1.79 | 1.22–2.64 | 0.003 |

OR = odds ratio; CI$_{95}$ = 95% confidence interval.

species tested in the present study, *C. coli* isolates were associated with an increased risk of resistance (OR 1.79; CI$_{95}$ 1.22–2.64) compared to *C. jejuni*. In a univariable analysis, QLD isolates were associated with a lower prevalence of resistance (OR 0.42; CI$_{95}$ 0.23–0.77), however the extent of this association was reduced in the multivariable analysis (OR 0.70; CI$_{95}$ 0.35–1.40). It should be noted that isolates were not collected in all sampling quarters in each region (S2 Fig).

## AMR prevalence by species

Nine genes and two mutations associated with antibiotic resistance were identified in the 331 *C. coli* isolates (Table 3; S3–S5 Figs). The *bla*$_{OXA-61}$ gene encoding resistance to β-lactams was the most commonly detected resistance gene, present on average in 80.4% of isolates (266/331). However, of the isolates possessing the *bla*$_{OXA-61}$ gene, none are predicted to have an active promotor, with all isolates having a G at position 57. Among *C. coli* isolates, only isolates from pork offal (10/28; 35.7%) contained the *lnu*(C) gene. The prevalence of several resistance genes, namely *aad9*, *lnu*(C) and *tet*(O), as well as the 23S rRNA A → G mutation at position 2075, were highest in isolates recovered from pork offal compared to isolates from other sources. The T86I mutation in the *gyrA* gene was present in isolates from all sources, except beef.

Ten genes and three mutations associated with antibiotic resistance were identified in the 285 *C. jejuni* isolates (Table 4; S5 Fig). The *bla*$_{OXA-61}$ gene was the most commonly detected antimicrobial resistance gene in *C. jejuni*, present on average in 79.0% of isolates (225/285). However, only 8.9% of these isolates (20/225) had an active *bla*$_{OXA-61}$ promoter (G → T mutation at position 57). The *tet*(O) gene was present on average in 13.3% (38/285) of *C. jejuni* isolates and was detected in isolates from all sources except beef offal. The T86I mutation in the protein encoded by the *gyrA* gene was detected infrequently (27/285; 9.5%) in *C. jejuni* isolates from beef, chicken and pork products. The A → G mutation in the 23S rRNA gene at position 2075 was only detected in one isolate from pork offal.

**Table 3. Summary of genetic determinants of antimicrobial resistance (AMR) present in *C. coli* recovered from beef, chicken, lamb and pork collected at retail outlets in the Australian Capital Territory, New South Wales, Queensland and Victoria, 2017–2019.**

| | No. of AMR genes and mutations (%[a]) | | | | | | | | | | | |
| | Aminoglycoside | | | | | β-lactam | | Lincosamide | Tetracycline | | Quinolone | Macrolide |
| Source | *aad9* | *aadE-Cc* | *ant(6)-Ia* | *ant(6)-Ib* | *aph(3')-IIIa* | *bla*$_{OXA-61}$ | *bla*$_{OXA-61}$ G57T | *lnu*(C) | *tet*(O) | *tet*(44) | *gyrA* T86I | 23 rRNA A2075G |
|---|---|---|---|---|---|---|---|---|---|---|---|---|
| **Beef** (*n* = 4) | 0 (0.0) | 1 (25.0) | 0 (0.0) | 0 (0.0) | 0 (0.0) | 4 (100.0) | 0 (0.0) | 0 (0.0) | 1 (25.0) | 0 (0.0) | 0 (0.0) | 0 (0.0) |
| **Chicken meat** (*n* = 190) | 0 (0.0) | 31 (16.3) | 0 (0.0) | 0 (0.0) | 0 (0.0) | 148 (77.9) | 0 (0.0) | 0 (0.0) | 21 (11.1) | 0 (0.0) | 15 (7.9) | 0 (0.0) |
| **Chicken offal** (*n* = 97) | 0 (0.0) | 15 (15.5) | 0 (0.0) | 0 (0.0) | 0 (0.0) | 86 (88.7) | 0 (0.0) | 0 (0.0) | 16 (16.5) | 0 (0.0) | 10 (10.2) | 1 (1.0) |
| **Lamb** (*n* = 12) | 0 (0.0) | 6 (50.0) | 0 (0.0) | 0 (0.0) | 0 (0.0) | 11 (91.7) | 0 (0.0) | 0 (0.0) | 2 (16.7) | 0 (0.0) | 3 (25.0) | 0 (0.0) |
| **Pork** (*n* = 28) | 8 (28.6) | 9 (32.1) | 1 (3.6) | 1 (3.6) | 1 (3.6) | 17 (60.7) | 0 (0.0) | 10 (35.7) | 15 (53.6) | 1 (3.6) | 1 (3.6) | 12 (42.9) |
| **Total** (*n* = 331) | 8 (2.4) | 62 (18.7) | 1 (0.3) | 1 (0.3) | 1 (0.3) | 266 (80.4)[b] | 0 (0.0) | 10 (3.0) | 55 (16.6) | 1 (0.3) | 29 (8.8) | 13 (3.9) |

[a]Values represent the percentage of *C. coli* isolates with the respective AMR gene or point mutation from each source.

[b] Phenotype is inferred as sensitive to ampicillin based on sequence at base 57 of the promotor.

## Phylogeny and AMR

Phylogenetically, *C. coli* isolates were highly diverse, forming two genetically-distinct clades, with isolates from pork offal generally more closely related to other pork isolates than those from other sources (S3–S5 Figs). Half the pork offal *C. coli* isolates had a multidrug resistance (MDR) genotype and among these, 57.1% (8 isolates) represented novel STs (S3 Fig). The T86I mutation in the protein encoded by the *gyrA* gene was restricted primarily to three STs: ST2083 (*n* = 10), ST7323 (*n* = 14) and ST860 (*n* = 24) (S3 & S5 Figs). In contrast, the *tet*(O) and *tet*(44) genes were identified in over 35 different STs (S3–S5 Figs). Fourteen isolates possessed the 23S rRNA A2075G mutation, representing 13 different STs. Multidrug resistance

**Table 4. Summary of genetic determinants of antimicrobial resistance (AMR) in *C. jejuni* recovered from beef, chicken, lamb and pork collected at retail outlets in the Australian Capital Territory, New South Wales, Queensland and Victoria, 2017–2019.**

| | No. of AMR genes and mutations (%[a]) | | | | | | | | | | Tetracycline | Quinolone | Macrolide |
| | β-lactam | | | | | | | | | | | | |
| Source | *bla*$_{OXA-184}$ | *bla*$_{OXA-185}$ | *bla*$_{OXA-449}$ | *bla*$_{OXA-460}$ | *bla*$_{OXA-466}$ | *bla*$_{OXA-61}$ | *bla*$_{OXA-61}$ G57T | *bla*$_{OXA-624}$ | *bla*$_{OXA-625}$ | *bla*$_{OXA-631}$ | *tet*(O) | *gyrA* T86I | 23 rRNA A2075G |
|---|---|---|---|---|---|---|---|---|---|---|---|---|---|
| **Beef** (*n* = 25) | 0 (0.0) | 0 (0.0) | 0 (0.0) | 0 (0.0) | 0 (0.0) | 24 (96.0) | 1 (4.0) | 0 (0.0) | 0 (0.0) | 0 (0.0) | 0 (0.0) | 2 (8.0) | 0 (0.0) |
| **Chicken meat** (*n* = 156) | 3 (1.9) | 1 (0.6) | 1 (0.6) | 1 (0.6) | 6 (3.9) | 110 (70.5) | 15 (9.6) | 1 (0.6) | 1 (0.6) | 1 (0.6) | 29 (18.6) | 18 (11.5) | 0 (0.0) |
| **Chicken offal** (*n* = 41) | 1 (2.4) | 0 (0.0) | 1 (2.4) | 0 (0.0) | 0 (0.0) | 31 (75.6) | 1 (2.4) | 0 (0.0) | 1 (2.4) | 0 (0.0) | 4 (9.8) | 3 (7.3) | 0 (0.0) |
| **Lamb** (*n* = 47) | 0 (0.0) | 0 (0.0) | 0 (0.0) | 0 (0.0) | 0 (0.0) | 46 (97.9) | 0 (0.0) | 0 (0.0) | 0 (0.0) | 0 (0.0) | 1 (2.1) | 0 (0.0) | 0 (0.0) |
| **Pork** (*n* = 16) | 1 (6.3) | 0 (0.0) | 0 (0.0) | 0 (0.0) | 0 (0.0) | 14 (87.5) | 3 (18.8) | 0 (0.0) | 0 (0.0) | 0 (0.0) | 4 (25.0) | 4 (25.0) | 1 (6.3) |
| **Total** (*n* = 285) | 5 (1.8) | 1 (0.4) | 2 (0.7) | 1 (0.4) | 6 (2.1) | 225 (79.0)[b] | 20 (7.0) | 1 (0.4) | 2 (0.7) | 1 (0.4) | 38 (13.3) | 27 (9.5) | 1 (0.4) |

[a]Values represent the percentage of *C. jejuni* isolates with the respective AMR gene or point mutation from each source.

[b]20/225 have an active promotor, inferring these isolates are phenotypically sensitive to ampicillin.

genotypes were found in 26 (7.6%) of *C. coli* and 2 (0.7%) *C. jejuni* (S3 & S5 Figs). *C. jejuni* isolates were highly diverse and generally did not cluster genetically by food source, except ST61 (S5 Fig). Among STs containing more than one isolate, only ST132 (*n* = 2), ST829 (*n* = 7), ST4896 (*n* = 13) and ST10123 (*n* = 2) had no resistance genes or mutations detected (S3–S5 Figs).

## Discussion

Antimicrobial resistance in enteric pathogens is considered a serious global public health problem. Among developed countries, several surveillance programs monitoring AMR of *Campylobacter* to clinically important antimicrobials have been well established, such as the U.S. National Antimicrobial Resistance Monitoring Systems [36], the Canadian Antimicrobial Resistance Surveillance System [37], the European Food Safety Authority [38] and the European Centre for Disease Prevention and Control [39]. In Australia, there is no national surveillance program for AMR monitoring in *Campylobacter* spp., limiting our knowledge of resistance in *Campylobacter*. The present study represents the first longitudinal genomic study of *Campylobacter* in meat at the retail level in Australia, providing insights into the prevalence of AMR genetic markers, as well as the genetic relatedness of isolates from different animal sources. Here we chose to collect samples at retail as this is the closest point before the product comes in contact with consumers, likely providing good representation of *Campylobacter* spp. that may cause illness in humans.

Although, we did not perform phenotypic antimicrobial susceptibility testing on the isolates in this study, our previous work [40] and reports by others [14, 28, 41], has shown a high concordance between resistance genotype and resistance phenotype in *Campylobacter*. Genetic evidence for resistance to tetracycline and fluoroquinolone antimicrobials was found in over 11 and 7.5% of *C. coli* isolates from chicken meat, respectively. These findings are comparable with those in a recent report published by the Australian Chicken Meat Federation (ACMF) that showed 5.2 and 3.1% of *C. coli* from chicken cecal samples were phenotypically resistant to fluoroquinolone and tetracycline, respectively [42]. Among *C. jejuni* isolates from retail chicken meat, genetic evidence for resistance to tetracycline and fluoroquinolone was 18.6 and 11.5%, respectively. These findings are slightly lower than those reported by the ACMF where 22.2 and 14.8% of *C. jejuni* isolates from chicken cecal samples were phenotypically resistant to tetracycline and ciprofloxacin, respectively [42].

Many of our results are understandable in the context of antibiotic usage patterns in Australian food-producing animals and recent literature on the evolution of resistance patterns. Possible explanations for the finding of mutations suggesting quinolone resistance in Australian food-producing animals, in the absence of quinolone usage, are a fitness advantage [43] in the absence of antibiotic selection, or co-selection [44] where found associated with other resistance patterns. The Australian poultry meat industry has suggested the initial introduction of fluoroquinolone resistance could have been from reverse zoonosis [42]. The moderate levels of aminoglycoside resistance could be due to usage of neomycin or spectinomycin rather than gentamycin, which is banned from use in food-producing animals. The range of mutations associated with AMR present in isolates from pork is reflective of the therapeutic usage of first-line antibiotics. Amoxycillin is commonly used to treat endemic respiratory and enteric diseases of pigs, including *Actinobacillus pleuroneumonia*, colibacillosis, enterotoxic *E. coli*, diarrhoea and Glasser's disease (*Haemophilus parasuis*). Tetracyclines are also commonly used to treat colibacillosis in pigs. Macrolides are commonly used in pigs to treat pneumonia due to mycoplasmosis, swine dysentery (*Brachyspira hyodysenteriae*) and proliferative enteropathy

(*Lawsonia intracellularis*). It is worth noting that there are no antibiotics used for growth promotion in Australian pigs.

Globally, resistance to clinically important antimicrobials is a serious threat to public health. A recent report by the European Food Safety Authority (EFSA) showed levels of AMR, particularly to tetracycline and ciprofloxacin, are very high in human and animal *Campylobacter* isolates in some European countries [45]. Prevalence of fluoroquinolone resistance in *Campylobacter* in Europe from food animals is highly variable, ranging from 1.2% in Norway [46], 44.0% in Belgium [47] and 82.9% in Italy [48] to 90.0% in Spain [49]. In Canada, ciprofloxacin resistance in *Campylobacter* isolated from retail chicken meat ranged from 6% in Québec to 35% in British Columbia [50]. In China, a recent report found the *tet*(O) gene in 98% of *Campylobacter* and the GyrA mutation at codon 86 and the 23S rRNA A2075G point mutation in 99 and 37% of isolates, respectively [51]. In China, antibiotic use is more than five times higher than the international average, primarily due to widespread misuse associated with growth promotion in feed and veterinary use on farms [52]. In Australia, the very low prevalence of AMR genetic markers in *Campylobacter* spp. from retail products is a testament to good antibiotic stewardship by the Australian agriculture industry. However, our findings suggest the prevalence of AMR in *Campylobacter* isolates from retail products may be increasing, highlighting the need for routine monitoring of resistance in *Campylobacter* in the agriculture industry. Genetic mutations conferring AMR may arise after continued exposure to antibiotics, but persist due to neutral selection, e.g. mutation in the *gyrA* gene conferring fluoroquinolones resistance. Although mutations in the 23S gene conferring erythromycin resistance are detrimental and are rapidly lost in the absence of selection. Future studies should examine resistance in *Campylobacter* spp. over an extended period of time to determine if the prevalence of resistance is increasing or if our findings can be regarded as background levels of resistance in Australian *Campylobacter*.

Campylobacteriosis is generally self-limiting and does not require medical treatment. However, people at increased risk of severe complications, such as the immunocompromised or elderly, will likely be prescribed antibiotics to prevent bacteraemia or sequelae [53]. This highlights the importance of monitoring prevalence of antimicrobial resistant *Campylobacter* in food. We identified five genes; *aad9*, *aadE-Cc*, *ant(6)-Ia*, *ant(6)-Ib* and *aph(3')-IIIa*, associated with resistance to aminoglycosides, as well as *lnu*(c), associated with lincomycin resistance, in *C. coli* but not *C. jejuni*. Additionally, the 23S rRNA mutation associated with resistance to macrolides was rare in both *Campylobacter* species, but more prevalent in *C. coli* than *C. jejuni*. Similarly, prevalence of MDR genotypes was ten-fold higher in *C. coli* than *C. jejuni*. In cases of campylobacteriosis in at risk individuals, macrolides or fluoroquinolones may be prescribed, with macrolides preferred due to their low rate of resistance. Our findings highlight that while a particular antimicrobial may be effective against *C. jejuni*, it may not be as effective for *C. coli*. In Australia, pathology laboratories do not routinely identify *Campylobacter* to species level. However, in cases where antibiotics are required, species identification may improve treatment success.

Australia is a large, geographically diverse, island nation supporting a distinct and diverse *Campylobacter* population evident in the findings from this study. Among the 616 *Campylobacter* isolates, representing 113 STs, 38 STs had not previously been reported anywhere else in the world. Many of the more common STs found in Australia, such as ST825, ST827 and ST50, have been isolated from various sources globally. *C. jejuni* isolates were dominated by ST50 which has been found in poultry in Israel [54], Poland [55], Denmark [56] and South Korea [57], among other countries. To date, ST50 has over 3,500 submissions on the PubMLST database. Unlike ST50 which was found in all animal sources, ST61 was dominated by isolates from non-poultry sources, namely beef and lamb offal. This finding agrees with other reports indicating ST61 is commonly recovered from ovine, bovine and human samples,

but rarely from poultry [58, 59]. There was no general association between different genetic markers of resistance and ST, with a few exceptions. Most isolates harbouring the GyrA T86I mutation, which is associated with resistance to ciprofloxacin, belonged to ST860, ST2083 or ST7323. Future studies should examine the prevalence of these STs in clinical isolates. The potential for the novel STs identified for the first time in the present study to cause illness in humans remains to be determined.

Of note, the $bla_{OXA-61}$ gene, shown to confer resistance to β-lactams [60], was present in 79.7% of *Campylobacter* isolates. However, rather than the presence of $bla_{OXA-61}$ alone, a single nucleotide mutation (G → T transversion) upstream of $bla_{OXA-61}$ has been shown to up-regulate expression of the gene, resulting in a high level of β-lactam resistance in *C. jejuni* [24]. In a previous study, we found 100% of *Campylobacter* isolates possessing the G → T mutation were resistant to ampicillin, while *C. jejuni* isolates possessing the $bla_{OXA-61}$ gene without this mutation, remained susceptible to ampicillin [40]. This information, taken with the other beta-lactamase genes we identified, suggests less than 15% of the *C. jejuni* isolates in the present study are likely to be resistant to β-lactam antibiotics. Beta-lactams are not recommended for treating campylobacteriosis as *Campylobacter* is intrinsically resistant to this class of antibiotic [48]. However, our findings suggest this intrinsic resistance may be less prevalent in *Campylobacter* from retail products in Australia, particularly in *C. jejuni*.

MDR is defined as resistance to three or more classes of antimicrobials. Among the 616 *Campylobacter* isolates examined in this study, 26 *C. coli* and two *C. jejuni* possessed MDR genotypes. However, among *C. coli* possessing a MDR genotype, more than 75% of these isolates had the $bla_{OXA-61}$ gene. Phenotypic testing is necessary to determine if the presence of the $bla_{OXA-61}$ gene alone in *C. coli* is in fact associated with phenotypic β-lactam antibiotic resistance. The majority of MDR *C. coli* represented STs observed for the first time in this study, and the role of these novel STs in human illness is unknown. Overall, our results show that MDR is very rare (< 5%) in Australia. This is comparable to findings in Poland where no MDR *Campylobacter* were found in poultry [61]. In contrast, resistance to multiple drugs (≥ 4) was found in 31.6% of *C. jejuni* isolated from poultry meat and related samples in Northern India [62]. Similarly, 8.6% of *C. jejuni* and 67.6% of *C. coli* isolates from diarrheal patients and poultry meat in Shanghai, China were identified as MDR [51]. Previous reports have indicated MDR *Campylobacter* can spread from the food supply chain into the human population [51]. In the present study, we noted differences in the prevalence of genetic determinants of resistance by food source. Generally, genes associated with resistance to aminoglycosides (except *aadE-Cc*), lincosamides and macrolides were only found in isolates from pork offal. Similarly, the majority (61.5%) of isolates harbouring MDR genotypes were recovered from pork offal. Although the prevalence of AMR is low in Australian *Campylobacter* spp., Australia should consider establishing integrated surveillance systems to monitor resistance and the transmission of *Campylobacter* from food to humans.

Our study has some limitations. Although we collected more than 700 samples from non-poultry sources, the majority of isolates that were sequenced (78.6%) were from chicken meat or chicken offal, due to the relatively low prevalence in beef, lamb and pork offal (14–38%) [18]. We may have achieved a higher recovery rate if we had sampled prior to slaughter rather than at retail as chilling significantly reduces the levels of *Campylobacter* [63]. Although Australians consume more muscle meat than offal, we also chose to collect offal samples of non-poultry products as the prevalence of *Campylobacter* has been reported to be higher in offal than muscle meat [64]. We found for the more common STs (> 10 isolates/ST) the *Campylobacter* isolates recovered from chicken meat or chicken offal products belonged to similar STs. As a result, the isolates recovered from offal likely provide a good representation of the STs that may be present on beef, lamb and pork meat.

## Conclusions

Our results indicate Australia's AMR prevalence in *Campylobacter* spp. from retail products is very low. Our results also suggest prevalence of resistance in *Campylobacter* spp. from foods of animal origin may be increasing, but ongoing surveillance is needed to confirm such a trend. Although consumption of contaminated poultry is well established as a key risk factor for campylobacteriosis, foods derived from other animals can result in *Campylobacter* infection. We found that isolates from pork represented a diverse array of STs, many not found among isolates from beef, chicken or lamb. MDR prevalence was also higher in isolates from pork offal, with many of these isolates representing novel STs. As these novel STs were first reported in this study, their significance in human health remains to be determined. Routine surveillance aimed at identifying species and characterizing types and resistance determinants from WGS data of *Campylobacter* spp. from food and humans will enable early detection of emergent AMR clones and ultimately assist in maintaining the low prevalence of AMR in Australia.

## Supporting information

**S1 Fig. Summary of *Campylobacter* isolates, from retail beef, chicken, lamb or pork products, by sampling quarter.**
(TIFF)

**S2 Fig. Summary of *Campylobacter* isolates, collected from retail products in the Australian Capital Territory (ACT), New South Wales (NSW), Queensland (QLD) and Victoria (VIC), by sampling quarter.**
(TIFF)

**S3 Fig. Phylogenetic analysis of *Campylobacter coli* group 1 (*n* = 317) isolated from retail beef, chicken, lamb or pork products.** The circle lanes from inner to outer order indicate: food source, number of antimicrobial resistance (AMR) genes or mutations, type of AMR gene or mutation (coloured by antibiotic class) and MLST number. Novel STs identified in this study are shown in red text.
(JPG)

**S4 Fig. Phylogenetic analysis of *Campylobacter coli* group 2 (*n* = 16) isolated from retail beef, chicken, lamb or pork products.** The lanes from inner to outer order indicate: food source, number of antimicrobial resistance (AMR) genes, type of AMR gene (coloured by antibiotic class) and MLST number. Novel STs identified in this study are shown in red text.
(JPG)

**S5 Fig. Phylogenetic analysis of *Campylobacter jejuni* (*n* = 286) isolated from retail beef, chicken, lamb or pork products.** The circle lanes from inner to outer order indicate: food source, number of antimicrobial resistance (AMR) genes or mutations, type of AMR gene or mutation (coloured by antibiotic class) and MLST number. Novel STs identified in this study are shown in red text.
(JPG)

**S1 Table. Accession numbers of *Campylobacter coli* and *Campylobacter jejuni* collected from retail beef, chicken, lamb and pork products from Australian retail outlets.**
(PDF)

## Acknowledgments

The CampySource Project Team comprises three working groups and a reference panel. The working groups focus on: food and animal sampling, epidemiology and modelling, and genomics. The reference panel includes expert representatives from government and industry. The study includes the following partner organisations: the Australian National University, Massey University, University of Melbourne, Queensland Health, Queensland Health Forensic and Scientific Services, New South Wales Food Authority, New South Wales Health, Hunter New England Health, Victorian Department of Health and Human Services, Food Standards Australia New Zealand, Commonwealth Department of Health and AgriFutures Australia–Chicken Meat Program. CampySource also collaborates with the following organisations: ACT Health, Sullivan Nicolaides Pathology, University of Queensland, Primary Industries and Regions South Australia, Department of Health and Human Services Tasmania, Meat and Livestock Australia, and New Zealand Ministry for Primary Industries. The CampySource Project Team consists of: Nigel P French, Massey University, New Zealand; Mary Valcanis, The University of Melbourne; Dieter Bulach, The University of Melbourne; Emily Fearnley, The Australian National University and South Australian Department for Health and Wellbeing; Russell Stafford, Queensland Health; Amy Jennison, Queensland Health; Trudy Graham, Queensland Health; Keira Glasgow, Health Protection NSW; Kirsty Hope, Health Protection NSW; Themy Saputra, NSW Food Authority; Craig Shadbolt, NSW Food Authority; Arie H Havelaar, The University of Florida, USA; Joy Gregory, Department of Health and Human Services, Victoria; James Flint, Hunter New England Health; Simon Firestone, The University of Melbourne; James Conlan, Food Standards Australia New Zealand; Ben Daughtry, Food Standards Australia New Zealand; James J Smith, Queensland Health; Heather Haines, Department of Health and Human Services, Victoria; Sally Symes, Department of Health and Human Services, Victoria; Barbara Butow, Food Standards Australia New Zealand; Liana Varrone, The University of Queensland; Linda Selvey, The University of Queensland; Tim Sloan-Gardner, ACT Health; Deborah Denehy, ACT Health; Radomir Krsteski, ACT Health; Natasha Waters, ACT Health; Kim Lilly, Hunter New England Health; Julie Collins, Hunter New England Health; Tony Merritt, Hunter New England Health; Rod Givney, Hunter New England Health; Joanne Barfield, Hunter New England Health; Ben Howden, The University of Melbourne; Kylie Hewson, AgriFutures Australia–Chicken Meat Program; Dani Cribb, The Australian National University; Rhiannon Wallace, The Australian National University; Angus McLure, The Australian National University; Ben Polkinghorne, The Australian National University; Cameron Moffatt, The Australian National University; Martyn Kirk, The Australian National University; and Kathryn Glass, The Australian National University. The authors thank Professor Glenn Browning (University of Melbourne) for critically reviewing the manuscript and providing comments on the use of antimicrobials in food-producing animals in Australia.

## Author Contributions

**Conceptualization:** Dieter M. Bulach, Amy V. Jennison, Martyn D. Kirk, Kathryn Glass.

**Data curation:** Rhiannon L. Wallace, Dieter M. Bulach, Amy V. Jennison, Mary Valcanis, Angus McLure, James J. Smith.

**Formal analysis:** Rhiannon L. Wallace, Dieter M. Bulach, Angus McLure.

**Funding acquisition:** Martyn D. Kirk, Kathryn Glass.

**Investigation:** Amy V. Jennison, Mary Valcanis, James J. Smith, Trudy Graham, Themy Saputra, Simon Firestone, Sally Symes, Natasha Waters, Anastasia Stylianopoulos.

**Methodology:** Dieter M. Bulach, Amy V. Jennison, Mary Valcanis, James J. Smith, Martyn D. Kirk, Kathryn Glass.

**Project administration:** Martyn D. Kirk, Kathryn Glass.

**Resources:** Amy V. Jennison, Mary Valcanis, James J. Smith, Trudy Graham, Themy Saputra, Simon Firestone, Sally Symes, Natasha Waters, Anastasia Stylianopoulos.

**Software:** Rhiannon L. Wallace, Dieter M. Bulach, Amy V. Jennison, Angus McLure.

**Supervision:** Martyn D. Kirk, Kathryn Glass.

**Validation:** Rhiannon L. Wallace, Dieter M. Bulach, Amy V. Jennison, Mary Valcanis.

**Visualization:** Rhiannon L. Wallace, Dieter M. Bulach.

**Writing – original draft:** Rhiannon L. Wallace.

**Writing – review & editing:** Rhiannon L. Wallace, Dieter M. Bulach, Amy V. Jennison, Mary Valcanis, Angus McLure, James J. Smith, Trudy Graham, Themy Saputra, Simon Firestone, Sally Symes, Natasha Waters, Anastasia Stylianopoulos, Martyn D. Kirk, Kathryn Glass.

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
