## [Decision Letter · Decision Letter 0]

17 Jun 2020

PONE-D-20-14286

Molecular characterization of Campylobacter spp. recovered from beef, chicken, lamb and pork products at retail in Australia

PLOS ONE

Dear Dr. Glass,

Thank you for submitting your manuscript to PLOS ONE. After careful consideration, we feel that it has merit but does not fully meet PLOS ONE’s publication criteria as it currently stands. Therefore, we invite you to submit a revised version of the manuscript that addresses the points raised during the review process.

We look forward to receiving your revised manuscript.

Kind regards,

Iddya Karunasagar

Academic Editor

PLOS ONE

Additional Editor Comments:

Some minor changes have been recommended.

Journal Requirements:

2. We note that you are reporting an analysis of a microarray, next-generation sequencing, or deep sequencing data set. PLOS requires that authors comply with field-specific standards for preparation, recording, and deposition of data in repositories appropriate to their field. Please upload these data to a stable, public repository (such as ArrayExpress, Gene Expression Omnibus (GEO), DNA Data Bank of Japan (DDBJ), NCBI GenBank, NCBI Sequence Read Archive, or EMBL Nucleotide Sequence Database (ENA)). In your revised cover letter, please provide the relevant accession numbers that may be used to access these data. For a full list of recommended repositories, see http://journals.plos.org/plosone/s/data-availability#loc-omics or http://journals.plos.org/plosone/s/data-availability#loc-sequencing.

"The CampySource Project Team comprises three working groups and a reference

panel. The working groups focus on: food and animal sampling, epidemiology and

modelling, and genomics. The reference panel includes expert representatives from

government and industry. The study is supported by the following partner organisations: the

Australian National University, Massey University, University of Melbourne, Queensland

Health, Queensland Health Forensic and Scientific Services, New South Wales Food

Authority, New South Wales Health, Hunter New England Health, Victorian Department of

Health and Human Services, Food Standards Australia New Zealand, Commonwealth

Department of Health and AgriFutures Australia–Chicken Meat Program. CampySource is

also supported by collaboration with the following organisations: ACT Health, Sullivan

Nicolaides Pathology, University of Queensland, Primary Industries and Regions South

Australia, Department of Health and Human Services Tasmania, Meat and Livestock

Australia, and New Zealand Ministry for Primary Industries. The CampySource Project Team

consists of: Nigel P French, Massey University, New Zealand; Mary Valcanis, The

University of Melbourne; Dieter Bulach, The University of Melbourne; Emily Fearnley, The

Australian National University and South Australian Department for Health and Wellbeing;

Russell Stafford, Queensland Health; Amy Jennison, Queensland Health; Trudy Graham, Queensland Health; Keira Glasgow, Health Protection NSW; Kirsty Hope, Health Protection

NSW; Themy Saputra, NSW Food Authority; Craig Shadbolt, NSW Food Authority; Arie H

 Havelaar, The University of Florida, USA; Joy Gregory, Department of Health and Human

 Services, Victoria; James Flint, Hunter New England Health; Simon Firestone, The

University of Melbourne; James Conlan, Food Standards Australia New Zealand; Ben

Daughtry, Food Standards Australia New Zealand; James J Smith, Queensland Health;

Heather Haines, Department of Health and Human Services, Victoria; Sally Symes,

Department of Health and Human Services, Victoria; Barbara Butow, Food Standards

Australia New Zealand; Liana Varrone, The University of Queensland; Linda Selvey, The

 University of Queensland; Tim Sloan-Gardner, ACT Health; Deborah Denehy, ACT Health;

Radomir Krsteski, ACT Health; Natasha Waters, ACT Health; Kim Lilly, Hunter New

 England Health; Julie Collins, Hunter New England Health; Tony Merritt, Hunter New

 England Health; Rod Givney, Hunter New England Health; Joanne Barfield, Hunter New

England Health; Ben Howden, The University of Melbourne; Kylie Hewson, AgriFutures

 Australia–Chicken Meat Program; Dani Cribb, The Australian National University; Rhiannon

 Wallace, The Australian National University; Angus McLure, The Australian National

 University; Ben Polkinghorne, The Australian National University; Cameron Moffatt, The

Australian National University; Ma 485 rtyn Kirk, The Australian National University; and

Kathryn Glass, The Australian National University."

"This work was funded by a National Health and Medical Research Council grant (NHMRC GNT1116294), AgriFutures, Australian Government Department of Health, Food Standards Australia New Zealand, New South Wales Food Authority and Queensland Health. The National Health and Medical Research Council provided research fellowship funding for Martyn D. Kirk (APP1145997).

Additionally, because some of your funding information pertains to commercial funding, we ask you to provide an updated Competing Interests statement, declaring all sources of commercial funding.

In your Competing Interests statement, please confirm that your commercial funding does not alter your adherence to PLOS ONE Editorial policies and criteria by including the following statement: "This does not alter our adherence to PLOS ONE policies on sharing data and materials.” as detailed online in our guide for authors  http://journals.plos.org/plosone/s/competing-interests.  If this statement is not true and your adherence to PLOS policies on sharing data and materials is altered, please explain how.

Please include the updated Competing Interests Statement and Funding Statement in your cover letter. We will change the online submission form on your behalf.

Reviewers' comments:

Reviewer's Responses to Questions

**Comments to the Author**

1. Is the manuscript technically sound, and do the data support the conclusions?

Reviewer #1: Yes

2. Has the statistical analysis been performed appropriately and rigorously? 

Reviewer #1: Yes

3. Have the authors made all data underlying the findings in their manuscript fully available?

Reviewer #1: Yes

4. Is the manuscript presented in an intelligible fashion and written in standard English?

Reviewer #1: Yes

5. Review Comments to the Author

Reviewer #1: I really liked reading the manuscript entitled “Molecular characterization of Campylobacter spp. recovered from beef, chicken, lamb and pork products at retail in Australia”. Here, the authors perform a molecular characterization using whole-genome sequencing of 616 Campylobacter spp isolates collected from different retail products in a two years period. The material, methods and results sections are very well written, and the discussion and conclusions are justified by the results obtained. Limitations are also well taken into consideration in the discussion section.

Minor comments:

- As in table 2, in Tables 3 and 4 it should be included in the legend the years of sample collection (2017-2019). The same in figure legends 1, 2 and 3. In this way the reader don´t miss the period time of the study at any moment of the paper.

- As a suggestion, “Results” section may be divided into subsections, for instance:

• Phylogeny

• AMR prevalence

• Presence of AMR genes and point mutations by species

• Phylogeny and AMR

6. PLOS authors have the option to publish the peer review history of their article (what does this mean?). If published, this will include your full peer review and any attached files.

Reviewer #1: No

---

## [Author Response · Author response to Decision Letter 0]

24 Jun 2020

Monday June 22nd, 2020

Dear editor and reviewers,

Thank you for taking the time to review our manuscript (PONE-D-20-14286) entitled " Molecular characterization of Campylobacter spp. recovered from beef, chicken, lamb and pork products at retail in Australia ". The comments have been very helpful in improving our manuscript. Comments from the Editor and Reviewer 1 are copied below (in black) and our responses are in blue. References to line numbers in the reponse to reviews are those in the final (unmarked) ‘Manuscript’.

Responses to the Editor’s recommendations:

Some minor changes have been recommended.

Journal Requirements:

The title page has been re-formatted to follow PLOS ONE’s style requirements. Specifically, the spacing on the title page and the author affiliations (components are now listed small to large) have been revised. All files uploaded as part of this revision have been named as suggested by the editor and include:

 ‘Response to Reviewers’

 ‘Revised Manuscript with Track Changes’

 ‘Manuscript’

Line 201- Changed “Table S1” to “S1 Table”

2. We note that you are reporting an analysis of a microarray, next-generation sequencing, or deep sequencing data set. PLOS requires that authors comply with field-specific standards for preparation, recording, and deposition of data in repositories appropriate to their field. Please upload these data to a stable, public repository (such as ArrayExpress, Gene Expression Omnibus (GEO), DNA Data Bank of Japan (DDBJ), NCBI GenBank, NCBI Sequence Read Archive, or EMBL Nucleotide Sequence Database (ENA)). In your revised cover letter, please provide the relevant accession numbers that may be used to access these data.

Next generation sequencing data has been deposited to NCBI’s SRA. Our sequence data is now publicly available. The reviewer link has been removed from line 202 and replaced with a link to the sequence data (https://www.ncbi.nlm.nih.gov/bioproject/PRJNA591966).

"The CampySource Project Team comprises three working groups and a reference

panel. The working groups focus on: food and animal sampling, epidemiology and

modelling, and genomics. The reference panel includes expert representatives from

government and industry. The study is supported by the following partner organisations: the

Australian National University, Massey University, University of Melbourne, Queensland

Health, Queensland Health Forensic and Scientific Services, New South Wales Food

Authority, New South Wales Health, Hunter New England Health, Victorian Department of

Health and Human Services, Food Standards Australia New Zealand, Commonwealth

Department of Health and AgriFutures Australia–Chicken Meat Program. CampySource is

also supported by collaboration with the following organisations: ACT Health, Sullivan

Nicolaides Pathology, University of Queensland, Primary Industries and Regions South

Australia, Department of Health and Human Services Tasmania, Meat and Livestock

Australia, and New Zealand Ministry for Primary Industries. The CampySource Project Team

consists of: Nigel P French, Massey University, New Zealand; Mary Valcanis, The

University of Melbourne; Dieter Bulach, The University of Melbourne; Emily Fearnley, The

Australian National University and South Australian Department for Health and Wellbeing;

Russell Stafford, Queensland Health; Amy Jennison, Queensland Health; Trudy Graham, Queensland Health; Keira Glasgow, Health Protection NSW; Kirsty Hope, Health Protection

NSW; Themy Saputra, NSW Food Authority; Craig Shadbolt, NSW Food Authority; Arie H

 Havelaar, The University of Florida, USA; Joy Gregory, Department of Health and Human

 Services, Victoria; James Flint, Hunter New England Health; Simon Firestone, The

University of Melbourne; James Conlan, Food Standards Australia New Zealand; Ben

Daughtry, Food Standards Australia New Zealand; James J Smith, Queensland Health;

Heather Haines, Department of Health and Human Services, Victoria; Sally Symes,

Department of Health and Human Services, Victoria; Barbara Butow, Food Standards

Australia New Zealand; Liana Varrone, The University of Queensland; Linda Selvey, The

 University of Queensland; Tim Sloan-Gardner, ACT Health; Deborah Denehy, ACT Health;

Radomir Krsteski, ACT Health; Natasha Waters, ACT Health; Kim Lilly, Hunter New

 England Health; Julie Collins, Hunter New England Health; Tony Merritt, Hunter New

 England Health; Rod Givney, Hunter New England Health; Joanne Barfield, Hunter New

England Health; Ben Howden, The University of Melbourne; Kylie Hewson, AgriFutures

 Australia–Chicken Meat Program; Dani Cribb, The Australian National University; Rhiannon Wallace, The Australian National University; Angus McLure, The Australian National University; Ben Polkinghorne, The Australian National University; Cameron Moffatt, The Australian National University; Martyn Kirk, The Australian National University; and Kathryn Glass, The Australian National University."

Please remove any funding-related text from the manuscript and let us know how you would like to update your Funding Statement. 

No funding information has been included in the acknowledgements section. Only those who have supported this study in the form of supervision, management, materials, resources and writing, or a combination of these, have been mentioned in the acknowledgements. The acknowledgements section has been revised (line 461) so that it is now clear that those mentioned in this section were partners. Some partner organizations provided a financial contribution, and these have been declared in the funding statement.

Currently, your Funding Statement reads as follows:

"This work was funded by a National Health and Medical Research Council grant (NHMRC GNT1116294), AgriFutures, Australian Government Department of Health, Food Standards Australia New Zealand, New South Wales Food Authority and Queensland Health. The National Health and Medical Research Council provided research fellowship funding for Martyn D. Kirk (APP1145997).

Additionally, because some of your funding information pertains to commercial funding, we ask you to provide an updated Competing Interests statement, declaring all sources of commercial funding.

We have updated our Competing Interests statement. The only source of commercial funding for this work was AgriFutures.

In your Competing Interests statement, please confirm that your commercial funding does not alter your adherence to PLOS ONE Editorial policies and criteria by including the following statement: "This does not alter our adherence to PLOS ONE policies on sharing data and materials.” as detailed online in our guide for authors http://journals.plos.org/plosone/s/competing-interests. If this statement is not true and your adherence to PLOS policies on sharing data and materials is altered, please explain how.

Please include the updated Competing Interests Statement and Funding Statement in your cover letter. 

Our updated Funding Statement can be found at the end of this letter, following our responses to the reviewer’s comments.

We will change the online submission form on your behalf.

The data being presented is not a core part of the research being presented in our study. Thus, both sentences have been removed.

Response to Reviewer #1 comments

Reviewer #1: I really liked reading the manuscript entitled “Molecular characterization of Campylobacter spp. recovered from beef, chicken, lamb and pork products at retail in Australia”. Here, the authors perform a molecular characterization using whole-genome sequencing of 616 Campylobacter spp isolates collected from different retail products in a two years period. The material, methods and results sections are very well written, and the discussion and conclusions are justified by the results obtained. Limitations are also well taken into consideration in the discussion section.

Minor comments:

- As in table 2, in Tables 3 and 4 it should be included in the legend the years of sample collection (2017-2019). The same in figure legends 1, 2 and 3. In this way the reader don´t miss the period time of the study at any moment of the paper.

Revised as suggested. The titles of Tables 1, 3 & 4 now include the years of sample collection.

- As a suggestion, “Results” section may be divided into subsections, for instance:

• Phylogeny

• AMR prevalence

• Presence of AMR genes and point mutations by species

• Phylogeny and AMR

As suggested, we divided the results section into the following subsections: ‘ST prevalence by source’ (line 206), ‘AMR prevalence over time’ (line 239), ‘AMR prevalence by species’ (line 264) and ‘Phylogeny and AMR’ (line 284).

Our updated Funding Statement is copied below.

This work was funded by a National Health and Medical Research Council grant (NHMRC GNT1116294), with partner funding from AgriFutures, Australian Government Department of Health, Food Standards Australia New Zealand, New South Wales Food Authority and Queensland Health. Additional funding was provided by ACT Health. The National Health and Medical Research Council provided research fellowship funding for Martyn D. Kirk (APP1145997).

Yours sincerely,

Rhiannon Wallace

Rhiannon.wallace@anu.edu.au

---

## [Decision Letter · Decision Letter 1]

16 Jul 2020

Molecular characterization of Campylobacter spp. recovered from beef, chicken, lamb and pork products at retail in Australia

PONE-D-20-14286R1

Dear Dr. Glass,

We’re pleased to inform you that your manuscript has been judged scientifically suitable for publication and will be formally accepted for publication once it meets all outstanding technical requirements.

Kind regards,

Iddya Karunasagar

Academic Editor

PLOS ONE

Additional Editor Comments (optional):

All reviewer comments have been addressed satisfactorily.

Reviewers' comments:

Reviewer's Responses to Questions

**Comments to the Author**

1. If the authors have adequately addressed your comments raised in a previous round of review and you feel that this manuscript is now acceptable for publication, you may indicate that here to bypass the “Comments to the Author” section, enter your conflict of interest statement in the “Confidential to Editor” section, and submit your "Accept" recommendation.

Reviewer #1: All comments have been addressed

2. Is the manuscript technically sound, and do the data support the conclusions?

Reviewer #1: (No Response)

3. Has the statistical analysis been performed appropriately and rigorously? 

Reviewer #1: (No Response)

4. Have the authors made all data underlying the findings in their manuscript fully available?

Reviewer #1: (No Response)

5. Is the manuscript presented in an intelligible fashion and written in standard English?

Reviewer #1: (No Response)

6. Review Comments to the Author

Reviewer #1: (No Response)

7. PLOS authors have the option to publish the peer review history of their article (what does this mean?). If published, this will include your full peer review and any attached files.

Reviewer #1: No

---

## [Editor Report · Acceptance letter]

20 Jul 2020

PONE-D-20-14286R1 

Molecular characterization of *Campylobacter* spp. recovered from beef, chicken, lamb and pork products at retail in Australia 

Dear Dr. Glass:

I'm pleased to inform you that your manuscript has been deemed suitable for publication in PLOS ONE. Congratulations! Your manuscript is now with our production department. 

Kind regards, 

on behalf of

Dr. Iddya Karunasagar 

Academic Editor

PLOS ONE